# Study on the Influence of Saturation on Freeze–Thaw Damage Characteristics of Sandstone

**DOI:** 10.3390/ma16062309

**Published:** 2023-03-13

**Authors:** Xinlei Zhang, Jiaxu Jin, Xiaoli Liu, Yukai Wang, Yahao Li

**Affiliations:** 1School of Civil Engineering, Liaoning Technical University, Fuxin 123000, China; 2Liaoning Key Laboratory of Mine Subsidence Disaster Prevention and Control, Fuxin 123000, China; 3State Key Laboratory of Hydroscience and Engineering, Tsinghua University, Beijing 100084, China

**Keywords:** freeze–thaw cycle, saturation, water migration, crack propagation rate, nuclear magnetic resonance

## Abstract

In order to explore the evolution mechanism of freeze–thaw disasters and the role of water in the freezing–thawing cycles of rocks, the macro mechanical indexes and microstructural characteristics of seven different saturation sandstones after certain freeze–thaw cycles were analyzed. Electron microscope scanning, nuclear magnetic resonance, and uniaxial compression tests were employed to study the migration law of water in the rock, the crack growth law, and the damage mechanism during freeze–thaw cycles. The results showed that when the saturation was 85%, the peak load curve of sandstone with different saturation appeared at the minimum point, and the porosity of sandstone reached the maximum. The damage variable increased sharply when the saturation was 75–85%. This proves that 85% saturation is the critical value of sandstone after five freeze–thaw cycles. The water migration freezing model is established, and the migration direction of capillary film water during freezing is micropore → mesopore → macropore. The migration of water is accompanied by the expansion and generation of cracks. Then we study the mechanism and law of crack expansion, and the crack propagation rate is positively related to the theoretical suction. The theoretical suction and theoretical ice pressure increased linearly with the decrease in temperature, which accelerated the crack propagation. The crack propagation rate in decreasing order is *V_macropore_* > *V_mesopore_* > *V_micropore_*. The research results can provide a theoretical basis for evaluating the stability of rocks under the action of freeze–thaw cycles in cold regions.

## 1. Introduction

According to statistics, the cold regions in China account for 43.5% of the land area. In recent years, the construction of geotechnical projects such as mines and tunnels in the western cold regions has gradually increased. Engineering geological and hydrogeological conditions are important influencing factors in many geotechnical engineering projects [1]. In these cold regions, the freeze–thaw cycle, accompanied by diurnal temperature variation and seasonal climate change, causes damage to and deterioration of the internal structure of the rock mass [2,3]. As a kind of easily damaged material, rock has many natural pores and intricate fractures in its interior. In addition, changes in the external environment, such as glacial water, groundwater, and rainwater, lead to the change in saturation that plays a decisive role in the mechanical properties of a rock mass under freeze–thaw cycles [4,5]. The damage and deterioration of the internal structure of a rock mass threaten the long-term stability of projects in cold regions. Therefore, for the safe operation and frost damage prevention of rock engineering in cold regions, it is of practical guiding significance to study the evolution law of freeze–thaw damage to sandstone with different initial saturations and the migration law of water molecules in the freezing process of sandstone.

The existing research has shown that the compressive strength, elastic modulus, Poisson’s ratio, and other mechanical properties of rock gradually decrease with the increase in the water content of rock in the process of freeze–thaw cycle [6,7], Hou et al. [8] found that the water chemical softening mechanism plays an important role in the process of rock freeze–thaw damage. Ren et al. [9] studied the change in mechanical properties and damage mechanism of saturated red sandstone under freeze–thaw and triaxial load. Ke et al. [10] found that the internal water content of rock aggravated the impact of freeze–thaw cycles on the dynamic mechanical properties of rock. Song et al. [11] analyzed the changes in mechanical properties, microstructure, and acoustic emission characteristics of sandstone with different water content under freezing and thawing. Lin et al. [12] found that freeze–thaw cycles led to the deterioration of the rock’s mechanical properties through crack propagation and evolution caused by frost pressure. Jia et al. [13] conducted freeze–thaw deformation monitoring tests on red sandstone with 50% saturation and 90% saturation and preliminarily explored the influence of saturation on rock freeze–thaw deformation. Rong et al. [14] studied the strength and micro damage mechanism of fissured yellow sandstone under freeze–thaw action and found that the freezing damage to single-jointed sandstone tended to form band damage with local fatigue characteristics, and the strain compliance and dissipated energy changed significantly under loading–unloading fatigue. Wang. et al. [15] conducted freeze–thaw cycle tests on saturated red sandstone and established attenuation models of uniaxial compressive modulus and deformation modulus considering the strain rate effect. Liu et al. [16] studied the uniaxial compressive strength and P-wave velocity of rocks with different water content under freeze–thaw action, and based on previous research results, they obtained an improved freeze–thaw damage model considering water saturation. Ma et al. [17] analyzed the energy evolution law of gypsum under different immersion times and established a constitutive damage model to describe the damage characteristics of gypsum rock under weak water effect and uniaxial compression based on energy dissipation. The damage to the mesostructure of rock under external load directly leads to the weakening of macro mechanical properties. Yang et al. [18] analyzed the microstructure damage to quartz sandstone under the coupling effect of water and freeze–thaw cycles using a scanning electron microscope [19]. Li et al. [20] obtained the T_2_ spectrum distribution and porosity variation of sandstone before and after freeze–thaw cycles using the nuclear magnetic resonance (NMR) technique. Liu et al. [21] used three-dimensional digital technology to study the change in the micropore structure of water-rich sandstone during freeze–thaw cycles and found that the freeze–thaw degradation mainly manifested in the increase in small pore size. Jia et al. [22] measured the T_2_ spectrum of sandstone during freeze–thaw process by the nuclear magnetic resonance (NMR) method and believed that there were two main reasons for freeze–thaw damage: one was that the water in the secondary crack migrated to the stem crack and froze, leading to the propagation and expansion of the stem crack; the other was that the unfrozen water in the secondary crack froze in situ, leading to the propagation, expansion, and generation of the secondary crack.

The above research has established a good foundation for correctly understanding the freeze–thaw damage characteristics of rocks with different initial saturation during freeze–thaw process. However, these research results mainly focused on macro mechanical characteristics or were limited to the meso freeze–thaw-damage mechanism. The research on the quantitative analysis of macro-meso variation characteristics has not been reported yet, and the damage mechanism of water molecules on rock fractures during freeze–thaw is still unclear. In view of this, this paper took seven kinds of sandstones with different initial saturation as the research objects and explored the changes of meso structural characteristics of sandstone with different saturation after freeze–thaw cycles. Furthermore, the macro mechanical characteristics under external load and the microstructure characteristics through electron microscopy scanning, nuclear magnetic resonance, and uniaxial compression tests were investigated. Based on the film water theory and capillary theory, a freeze–thaw damage model suitable for the whole freeze–thaw process was established.

## 2. Experimental

### 2.1. Sample Preparation

The samples for this test were taken from a rock slope in the transition section between Sichuan Basin and Qinghai-Tibet Plateau. According to the test procedures of the International Society of Rock Mechanics, the sandstone with high homogeneity, small dispersion, and large porosity was selected to make international standard rock samples (Ф50 mm × 100 mm). The parallelism and perpendicularity of both ends of the sample met the specification requirements, as shown in Figure 1. After the completion of the processing of the specimen, the defective and incomplete test pieces should be removed. The ultrasonic detector, which was produced by Beijing Gaotiejian Technology Development Co., Ltd. in Beijing, China, was used to test the sound time and sound velocity of the specimen with uniform texture and complete appearance. After the test, the specimens with large dispersion were removed, and 35 rock samples with similar quality should be selected as the test rock samples. The physical properties of the sandstone samples are summarized in Table 1. The samples were divided into seven groups; each group was numbered A1–A5, B1–B2, C1–C5, D1–D5, E1–E5, F1–F5, and G1–G5. The letters A–E correspond to 0%, 30%, 60%, 75%, 85%, 95%, and 100% saturation, respectively. According to the pre-test results, the mass change of the sample after drying at 105 °C for 48 h did not exceed 0.1%, which can be considered to indicate that the water content of the rock sample was 0%. So, the rock samples were dried in an oven at 105 °C for 48 h to obtain samples with 0% saturation. Samples with 30%, 60%, 75%, 85%, and 95% saturation were then prepared by using the chemical thermodynamic method. According to the “Specifications for rock tests in water conservancy and hydroelectric engineering,” the sample must be saturated using the vacuum pumping method. The pumping vacuum degree needs to reach 100 kPa negative pressure, and the pumping time is not less than 4 h. Therefore, the saturated rock samples were prepared in a vacuum saturation device (−0.1 Mpa) and taken out after 12 h and surface dried.

### 2.2. Test Process

According to the actual working conditions in cold regions, the freezing and melting temperatures were set to −20 °C and 20 °C, respectively, the cooling and freezing time lasted for 2 h and 10 h, respectively, and the time period of the rising and constant melting temperatures was 1 h and 11 h respectively. Each freeze–thaw cycle lasted for 24 h for five cycles. A JSM-7500F scanning electron microscope (Figure 2), which was produced by JEOL Ltd. in Tokyo, Japan, was used to observe the changes in the internal pore structure of sandstone with different saturation under freeze–thaw action, and a MesoMR23-060H-I nuclear magnetic resonance tester (Figure 3), which was produced by Shanghai Electronic Technology Co., Ltd. in Shanghai, China, was used to scan the sandstone after freeze–thaw action. The uniaxial compression test of the specimens after freeze–thaw treatment was carried out using a TAW-2000 electro-hydraulic servo rock triaxial tester which was produced by Changchun City Chaoyang Test Instrument Co., Ltd. in Changchun, China. The displacement loading method was used in the loading process with a loading rate of 0.12 mm/min.

## 3. Physical and Mechanical Properties of Sandstone under Freeze–Thaw Action

### 3.1. Analysis of Mechanical Properties

The uniaxial compression test on sandstone with different saturations was carried out by using a uniaxial press, and the curves of the uniaxial compression load versus displacement are in Figure 4. The change trend for displacement load curves of rock samples with different saturations is basically the same. The curves first rise to the peak value and then fall in steps. According to the load-displacement curves of sandstones, the uniaxial compression process of rock samples is divided into five stages: concave stage (OA), linear stage (AB), convex stage (BC), failure stage (CD), and post-failure deformation stage (DE).

With the increase in saturation, the concave stage is obviously prolonged under a certain axial pressure. This may be because, for samples with high water content, the hydraulic pressure compensated part of the axial pressure, and thus the closure time of initial microcracks and pores was prolonged. In the linear stage, when the saturation of rock increases, the slope of the curve decreases first and then increases, and the linear stage becomes shorter, which means that the higher saturation results in more severe deterioration after freeze–thaw cycles. This is because the damage caused by water–ice phase change during freeze–thaw cycles is intensified when the rock contains more water, which consequently induces more microcracks and weakens the particle composition connection between the rock samples. In the upward convex stage, the rock sample enters the nonlinear deformation stage, generating irreversible deformation. The duration of the upward convex stage shows a trend of increasing first and then decreasing with the increase in saturation. This is because, in this stage, the early microcracks and newly generated cracks converge to form a through-crack. The more microcracks in the rock sample, the longer the duration of this failure process will last. Relevant research results also verify this point [23]: the slope of the stress-strain curve of the saturated sample is smaller than that of the dry sample, and the accelerated crack growth stage of the saturated rock is shorter. In the failure stage, the peak strength decreases first and then increases as the saturation increases. It has also been stated in previous literature that with the increase in water content, the stress–strain curve becomes gentler, so water leads to a decrease in the strength and stiffness of red sandstone [16]. This is due to the significant development of cracks and the significant reduction in axial stress, which will produce more cracks that connect to form through cracks, and more pores will collapse as a result. Compared with dry samples, water-containing samples have a higher deformation rate, larger deformation amount, and more obvious crack propagation at this stage [24].

The slope of the deformation stage after failure gradually decreases with the increase in saturation. In the presence of excessive pore pressure, the particles in the cementitious material may fall off or break. These particles can be rearranged to form a new structural stress.

The linear slope of the stress–strain curve of sandstone with different saturation is taken as the elastic modulus, which approximately corresponds to 40–60% of the peak compressive stress. Table 2 is the uniaxial compression test results. According to the strain equivalence hypothesis proposed by Lemaitre [25], the internal freeze–thaw damage to the constitutive relationship of rock materials is defined as:(1)$$E_{n}=E_{0}(1-D_{n})$$
(2)$$\sigma_{n}=E_{0}(1-D_{n})\varepsilon_{n}$$
where $E_{0}$ and $E_{n}$ are the elastic moduli of the initial damage state and freeze–thaw damage state.

The peak load loss rate is defined as:(3)$$W_{n}=\frac{\left|\Delta F\right|}{F_{0}}\times 100\%=\frac{\left|F_{n}-F_{0}\right|}{F_{0}}\times 100\%$$
where $F_{n}$ is the peak load when the saturation is *n*, and $F_{0}$ is the peak load in the dry state.

Figure 5 shows the relationship between the peak load and damage variable with saturation. It is obviously found that there are two dividing points in the curve. When the saturation is less than 75%, the peak load loss rate is 24.1%, but when the saturation exceeds 75%, the peak load loss rate rises sharply, reaching 51.9%. Therefore, 75% is taken as a dividing point where the freezing crack threshold of pores is reached. When the saturation reaches about 85%, the peak load reaches the minimum value, and there is a slight increase afterward. The saturation threshold is considered to have been reached at this time.

### 3.2. Mesostructure Analysis

The microstructural characteristics of sandstones with different saturations after five freeze–thaw cycles were observed using a JSM-7500F scanning electron microscope, and the results are shown in Figure 6 and Figure 7. In the dry state, the internal structure of the sample is complete, the particles are closely arranged, the crack edges and corners are clear, the grain is clear, and the particle surface is smooth. There is no softening failure phenomenon or obvious debris. When the saturation is 30–85%, the destruction of the sandstone’s internal structure becomes more prominent. Specifically, the particles begin to soften and decompose, and the intra-particle pore volume increases correspondingly, resulting in a looser and more porous structure. When the saturation reaches 85–100%, the angularity of the crack boundary tends to be smooth, the local damage to the internal structure becomes more severe, the integrity becomes worse, and the cementation between particles weakens. With the increase in saturation, obvious brittle cracks appear between mineral particles, and the original internal gaps begin to expand and gradually converge.

According to the morphology analysis, water has a detrimental effect on rock sample structure. It mainly leads to a loose internal structure by dissolving rock cement, as sandstone cement is more soluble than quartz particles. The loss of cement results in a decrease in cement cohesion. Previous researchers have a similar view that the weakening effect of water on rock was divided into physical softening and chemical erosion. After long-term freeze–thaw cycles, rock constituent materials, which were mainly calcite in the studied specimens, dissolved in water [26,27]. On the other hand, the freeze–thaw damage to rocks is mainly caused by the volume expansion of 9% frozen water in pores and the thermal stress at low-temperature conditions. Due to the lack of free expansion space, the water in the sandstone pores produces a large frost-heaving force during the freezing process. Due to the different initial saturations, the formation and development degree of ice and the frost heave force of water are different under freezing conditions, and the degree of frost heave damage in the internal structure of rock samples is different [28]. When the saturation is low, the damage effect of water on rock is mainly chemical softening, and the frost-heaving effect of water is weak, so the damage effect of water on rock is limited. When the saturation exceeds 75%, the volume expansion of fissure ice causes the initiation, development, and expansion of new fissures, and the porosity increases rapidly. This trend continues until the saturation reaches 85% and the fracture expansion in sandstone reaches its peak. Subsequently, the interior of the sandstone gradually formed a penetration surface, and the integrity of the rock sample was destroyed. Therefore, the saturation of 85% is the critical value of sandstone damage deterioration after five freeze–thaw cycles.

### 3.3. Nuclear Magnetic Resonance Characteristic Analysis

Based on NMR experiments on sandstones with different saturations, the causes of frost crack threshold and saturation threshold of sandstones during freeze–thaw were explored. Figure 8 shows the T_2_ distribution of NMR of sandstones with different saturations after freeze–thaw cycles. On the whole, the maximum pore diameter shows a trend of first increasing and then stabilizing with the increase in saturation, while the proportion of micropores is just the opposite. The peak area of the T_2_ spectrum of sandstone with 85% saturation is the largest, which means that with the increase in saturation, the freezing process produces new pores in these samples, especially the macropores. This suggests that the samples are seriously damaged. Macropores have a trend of further expansion, resulting in the maximum relaxation time continuing to increase. Moreover, more peaks in the T_2_ characteristic curve are shown for saturated rocks, indicating more larger-sized pores are produced after freeze–thaw cycles. Although the freezing process does not cause macroscopic damage to rock samples, it leads to the continuous expansion of internal cracks in sandstone [29].

Figure 9 and Figure 10 show the pore size distribution derived from NMR results. As the saturation increases from 60% to 85%, the cumulative porosity increases significantly. However, as the saturation further increases to 100%, the cumulative porosity decreases to some extent. It shows that the porosity of the rock is greatly affected by saturation. The higher porosity facilitates fracture generation and expansion. With the saturation reaching 85%, the change trend for porosity gradually slows down, but the overall change trend is increasing.

Table 3 shows the NMR parameters of sandstone with different saturations. It can be found that when the saturation increases from 75% to 85%, the porosity of sandstone after freeze–thaw cycles increases by 15.67%, indicating that when the saturation is greater than 75%, the internal structure of sandstone is seriously damaged due to freeze–thaw action. This can be confirmed by the increase in rock permeability. When the saturation reaches 85%, the degree of damage to the internal structure of sandstone reaches the peak value, and the maximum porosity reaches 8.03%. With a further increase in saturation, it shows a small downward trend and then tends to be flat. The bound fluid increases first and then decreases to a stable value with the increase in saturation. In the process of saturation transition from 75% to 85%, the mutual conversion rate of bound fluid and free fluid is the largest, which also demonstrates that the damage to sandstone is the most severe. When the saturation reaches 85%, the bound fluid and permeability reach the peak value, making more free fluid absorbed by rock particles.

## 4. Damage Evolution Law of Sandstone under Freeze–Thaw Action

### 4.1. Water Migration Freezing Model

According to the size, shape, and connectivity of pores, they are divided into trunk holes, connecting holes, end closed holes, and closed holes. As shown in Figure 11, the main pore has a large size, so it is the main migration channel for pore water. The holes connected with the main hole are called side branch holes. The difference between the connecting hole and the end closed hole is that one end of the end closed hole is closed. The closed hole is independent of the main hole and the side branch hole and is not connected with other holes. During the freezing process, different pore types may be converted. The connected hole may become an end closed hole after one end is frozen, the end closed hole may become a closed hole after the main hole is frozen, and the closed hole may become an end closed hole due to the expansion of the pore.

According to previous studies, the heat conduction process of any material follows a gradient mode. Similarly, during the freezing process of sandstone, its internal temperature field changes unevenly, so it is necessary to consider the temperature transmission process when distinguishing the freezing conditions that the pore water meets in different freezing stages. According to solidification theory [30], the freezing rate of water is slow during the freezing process, so the process of the water–ice phase transition can be regarded as a static process. Moreover, there is a supercooling stage in the freezing process; that is, in the process of ice growth, a stable temperature field is formed in the cross-section of cylindrical fractures. The temperature of the environment$\;T_{f}$ where the rock is located is the same as the temperature of the boundary $R_{1}$, and in the pores, the temperature of the water/ice interface is the same as the temperature of the ice body, namely:(4)$$T=T_{f}\;\;\;\;\;\;\;\;r=R_{1}\gg R\left(t\right)$$
(5)$$T^{1}=T^{i}=T^{I}\;\;\;\;\;\;\;\;\;\;\;\;\;\;r\leq a$$

According to Laplace’s equation:(6)$$\nabla^{2}T=0$$

In polar coordinates, the two-dimensional Laplace equation is:(7)$$\frac{1}{r}\frac{\partial }{\partial r}\left(r\frac{\partial T}{\partial r}\right)=0$$

The general solution of Equation (7) can be expressed as T=i+jlnr, *i* and *j* are assumed to be constant. The temperature field distribution around the crack can be obtained by bringing Equations (4) and (5) into Equation (7):(8)$$T=T^{I}\;\;\;\;\;\;\;\;r\leq a$$
(9)$$T=T_{f}+\frac{T^{I}-T_{f}}{\ln\frac{a}{R_{1}}}\ln\frac{r}{R_{1}}\;\;\;\;\;\;\;\;r>a$$

According to the principle of energy conservation, the heat balance equation is:(10)$$l\frac{dm_{i}}{dt}=\left[\lambda_{T}^{i}\nabla T^{I}-\lambda_{T}^{1}\nabla T^{1}\right]s\cdot \vec{n}$$

By substituting Equations (8) and (9) into Equation (10), the relationship between the water–ice phase transition velocity and temperature field is obtained.
(11)$$l\rho_{i}a\dot{a}=\lambda_{T}^{1}\frac{T_{f}-T^{I}}{\ln\frac{a}{R_{1}}}$$
where l—latent heat of phase change, $\lambda_{T}^{i}$—thermal conductivity in ice, $\lambda_{T}^{1}$—thermal conductivity in unfrozen water, $m_{i}$—the quality of fissure ice, s—water-ice interface area,  n ⇀-outer normal direction of water–ice interface, $\dot{a}$—the growth rate of ice in the fissure, $\dot{a}=\frac{da}{dt}$, and $\rho_{i}$—Fissure water density.

With the decrease in pore radius, the freezing temperature also decreases. Therefore, the freezing direction of sandstone is consistent with the decreasing direction of pore size. Unfrozen water in different pores is classified according to the pore classification standard proposed by previous researchers [31] (Table 4).

According to the theory of thin film water and capillary theory, the influence of pore radius on the migration of unfrozen water in the freezing process was explored, and the ‘main-branch’ pore structure with poor frost resistance was analyzed. The ‘main-branch’ pore structure is composed of main pores, secondary pores, and micropores. Due to the good connectivity between pores and the small resistance of water migration, the frost resistance is poor. Based on the freezing temperature of the main pore, secondary pore, and micropore, the capillary film moisture transfer unit model was established in turn according to the freezing sequence, as shown in Figure 12.

It can be seen from Equation (11) that the freezing order is in the order of macropores, mesopores, and micropores, and the stress analysis is performed on the frozen state. The total suction is the source of hydraulic and gravitational acceleration and, in equilibrium, is
(12)$$P_{SU}=-\frac{v_{s}}{v_{l}}P_{S0}$$

The relationship between total suction and hydraulic pressure is
(13)$$P_{SU}=-\frac{v_{s}}{v_{l}}\left(P_{Lh}-\lambda \bar{K}\right)$$

(1)Macropore freezing

Assuming that the freezing temperature of the main hole is $T=T_{1}$, as shown in Figure 12a, the theoretical ice pressure at this time is
(14)$$P_{S1}=\frac{l\left(-T_{1}\right)}{v_{s}T_{a}}$$

According to capillary theory, capillary suction is
(15)$$P_{S1}=P_{C1}=\frac{2\gamma }{R_{1}}$$

According to the freezing temperature equation of pore water, the freezing temperature is
(16)$$T_{1}=-\frac{v_{s}T_{a}}{l}\frac{2\gamma_{sl}}{R_{1}}$$

According to the ice pressure, hydraulic, and interface force balance, the hydraulic pressure is the difference between the ice pressure and the interface pressure is
(17)$$P_{Lh1.1}=P_{S1}-\frac{\gamma }{R_{1}}=\frac{\gamma }{R_{1}}$$

From Equation (13), the total suction is
(18)$$P_{SU1.1}=-\frac{v_{s}}{v_{l}}\left(\frac{\gamma }{R_{1}}-\lambda \bar{K}\right)$$

At the initial stage of freezing, the ice pressure is greater than the hydraulic pressure; that is, the water–ice phase change speed is greater than the migration speed of capillary water and film water so that the unfrozen water will flow to other pores, causing a freezing overflow.

(2)Mesoporous freezing

When the temperature reaches $T=T_{2}$, as shown in Figure 12b, according to the capillary theory, the capillary suction is
(19)$$P_{S2}=P_{C2}=\frac{2\gamma }{R_{2}}$$

The hydraulic pressure of film water in macropores and mesopores is
(20)$$P_{Lh2.1}=\frac{2\gamma }{R_{2}}-\frac{\gamma }{R_{1}}$$
(21)$$P_{Lh2.2}=\frac{2\gamma }{R_{2}}-\frac{\gamma }{R_{2}}=\frac{\gamma }{R_{2}}$$

From Equation (13), the total suction is
(22)$$P_{SU2.1}=-\frac{v_{s}}{v_{l}}\left(\frac{2\gamma }{R_{2}}-\frac{\gamma }{R_{1}}-\lambda \bar{K}\right)$$
(23)$$P_{SU2.2}=-\frac{v_{s}}{v_{l}}\left(\frac{\gamma }{R_{2}}-\lambda \bar{K}\right)$$

It can be seen from $P_{SU2.1}<P_{SU2.2}$ that the total suction of macropores is stronger than that of mesopores, so unfrozen water migrates from micropores and mesopores to macropores.

(3)Micropore freezing

When the temperature reaches $T=T_{3}$, as shown in Figure 12c, according to the capillary theory, the capillary suction is
(24)$$P_{S2}=P_{C2}=\frac{2\gamma }{R_{2}}$$

The hydraulic pressure of thin film water in macropores, mesopores, and micropores is:(25)$$P_{Lh3.1}=\frac{2\gamma }{R_{3}}-\frac{\gamma }{R_{1}}$$
(26)$$P_{Lh3.2}=\frac{2\gamma }{R_{3}}-\frac{\gamma }{R_{2}}$$
(27)$$P_{Lh3.3}=\frac{2\gamma }{R_{3}}-\frac{\gamma }{R_{3}}=\frac{\gamma }{R_{3}}$$

From Equation (13), the total suction is:(28)$$P_{SU3.1}=-\frac{v_{s}}{v_{l}}\left(\frac{2\gamma }{R_{3}}-\frac{\gamma }{R_{1}}-\lambda \bar{K}\right)$$
(29)$$P_{SU3.2}=-\frac{v_{s}}{v_{l}}\left(\frac{2\gamma }{R_{3}}-\frac{\gamma }{R_{2}}-\lambda \bar{K}\right)$$
(30)$$P_{SU3.3}=-\frac{v_{s}}{v_{l}}\left(\frac{\gamma }{R_{3}}-\lambda \bar{K}\right)$$

It can be seen from $P_{SU3.1}<P_{SU3.2}<P_{SU3.3}$ that a macropore has the strongest adsorption capacity. The larger the pore diameter, the smaller the membrane water pressure. Therefore, it continues to drive the membrane water in mesopores and micropores to migrate to macropores. To sum up, the migration direction of capillary film water during freezing is micropore → mesopore → macropore.

### 4.2. Analysis of Crack Propagation Mechanism

We define the crack in the rock as a “coin” shape; that is, it is circular in plain view and elliptical in side view, with a radius of c, and has a very small opening. From the maximum value w in the center to 0 at the edge, it is assumed that the crack spacings are wide enough, and they grow independently of each other. The growth of all cracks is in the crack plane, which is completely caused by uniform internal ice pressure; that is, all crack growth is Type I. Under these conditions, Sneddon and Lowengrub [32] believed that crack growth in brittle elastic solids was controlled by the magnitude of stress intensity factors$\;K_{I}$.
(31)KI=4cπ12Ps

The film water in the freezing trunk hole is subjected to the maximum hydraulic pressure and total suction. The total suction will drive the water to move towards the freezing main hole continuously. Based on the capillary theory, the increase in hydraulic pressure will also lead to an increase in stress intensity factor, namely:(32)KI=4cπ12Ps

When $K_{I}$ exceeds the ‘fracture toughness’ value $K_{c}$ of the material, the crack propagates unstably at a speed close to the compressive elastic wave. When $K_{I}<K_{c}$, the crack propagation occurs slowly and stably. Segall [33] believed that when$\;K_{I}$ dropped below a critical value $K_{*}$, crack propagation should stop. There are many factors affecting the crack growth rate, such as temperature, pressure, and chemical environment. Considering water ice as a pure medium, Segall proposed an empirical formula for the crack growth rate:(33)$$V=V_{C}\left[e^{\varepsilon \left(\frac{K_{I}^{2}}{K_{c}^{2}}-1\right)}-e^{\varepsilon \left(\frac{K_{*}^{2}}{K_{c}^{2}}-1\right)}\right],K_{I}>K_{*}$$
(34)$$V=0,K_{I}\leq K_{*}$$
where V for crack propagation speed, $V_{C}$ and ε for material properties.

When the stress intensity factor $K_{I}$ is greater than the stress corrosion limit value *K_⁎_*, the pore ice will overcome the bondage of the pipe wall, crack, and expand at the weak points at both ends of the main hole, eventually leading to the destruction of the pore structure of the main hole and the increase in the fracture surface.

In the initial stage of freezing, the temperature gradient increases with the decrease in the radius of the rock sample cross-section; that is, the initial freezing process occurs on the rock surface, and therefore the unfrozen water in the rock will continue to migrate under the effect of the temperature gradient. With the intensification of the freezing process, the expansion of the frozen area and the continuous growth of ice would lead to the occurrence of freezing damage. At this time, the freezing environment meets the fractional ice theory in rock mass established by Walder et al. [34].

Gilpin [35] learned from the generalized Darcy’s law and the conservation of frozen water mass that the volume change of ice during the water/ice phase transition is
(35)$$V_{s}=\left(\frac{v_{s}^{2}}{gv_{l}}\right)\left(\frac{1}{R_{f}}\right)\left[\frac{l\left(-T_{r}\right)}{v_{s}T_{a}}-p_{s}+P_{Lf}\right]\cos\theta$$

If the fissure volume expands, the free water in the rock will migrate to the fissure under the combined action of gravity, capillary force, and membrane force, and the ice pressure will decrease rapidly. The change of water mass caused by the water–ice phase change in the pore is defined as
(36)$$\frac{dM_{r}}{dt}=\frac{\pi c^{2}V_{s}}{v_{s}}$$

By substituting Equation (35) into Equation (36), it can get:(37)$$\frac{dM_{r}}{dt}=\left(\frac{\pi c^{2}v_{s}}{gv_{l}}\right)\left(\frac{1}{R_{f}}\right)\left[\frac{l\left(-T_{r}\right)}{v_{s}T_{a}}-p_{s}+P_{Lf}\right]\cos\theta$$

Under the action of pressure, temperature, gravity, and other migration forces, the internal unfrozen water always migrates to the external frozen area so that the external pores will eventually be saturated. It is assumed that the pores are ellipsoidal, and the volume of pores is $\frac{2\pi wc^{2}}{3}$. Therefore, another manifestation of water quality change in pores is:(38)$$\frac{dM_{r}}{dt}=\left(\frac{2\pi }{3v_{s}}\right)\left(c^{2}\frac{dw}{dt}+2wc\frac{dc}{dt}\right)$$

It can be known from Equations (37) and (38) that the change rate of the fissure is
(39)$$\frac{dw}{dt}=\left(\frac{3v_{s}}{2gv_{l}}\right)\left(\frac{1}{R_{f}}\right)\left[\frac{l\left(-T_{r}\right)}{v_{s}T_{a}}-p_{s}\right]\cos\theta -2\left(\frac{w}{c}\right)\left(\frac{dc}{dt}\right)$$

The formula of freezing temperature changing with pore size is
(40)$$T_{r}=-\frac{v_{s}T_{a}}{l}\frac{2\gamma_{sl}}{R}$$

In the formula, $\gamma_{sl}$ is the interfacial tension of water/ice. Substituting Equation (40) into Equation (39) yields:(41)$$\frac{dw}{dt}=\left(\frac{3v_{s}}{2gv_{l}}\right)\left(\frac{1}{R_{f}}\right)\left[\frac{l}{v_{s}T_{a}}\left(\frac{v_{s}T_{a}}{l}\frac{2\gamma_{sl}}{R}\right)-p_{s}\right]\cos\theta -2\left(\frac{w}{c}\right)\left(\frac{dc}{dt}\right)$$
where w,  c,  θ are pore parameters. Because w and c represent the size of the crack, Formula (41) is simplified by substituting *R* for w and c:(42)$$\frac{dR}{dt}=\left(\frac{v_{s}}{gv_{l}}\right)\left(\frac{1}{R_{f}}\right)\left[\frac{\gamma_{sl}}{R}-p_{s}\right]\cos\theta$$

Because the hydraulic $P_{Lh}$ is
(43)$$P_{Lh}=\frac{\gamma_{sl}}{R}-p_{s}$$

Taking Equation (43) into Equation (42), the crack propagation rate is
(44)$$\frac{dR}{dt}=\left(\frac{v_{s}}{gv_{l}}\right)\left(\frac{1}{R_{f}}\right)P_{Lh}\cos\theta$$

According to Formula (44), the crack propagation rate is positively correlated with the theoretical suction. During the freezing process, the theoretical suction and the theoretical ice pressure increase with the decrease in temperature, which accelerates crack propagation. According to the division of macropores, mesopores, and micropores, the crack propagation rate is *V*_macropore_ > *V*_mesopore_ > *V*_micropore_.

### 4.3. Model Validation

The water in the rock sample is frozen into ice during the freezing process, and the volume expansion exerts pressure on the pore wall, leading to the growth of the pore structure and the formation of new pores in the rock. Figure 13 shows the relaxation time decay curve of porous media.

The migration of water molecules during the freezing process is studied based on the analysis of porosity, *T_2_* distribution, and NMR images. The transverse relaxation time *T_2_* of pore fluid is proportional to the pore size. The peak value and area of *T_2_* distribution represent the concentration degree and the number of pores with different sizes. In order to ensure the reference of the test, a group of saturated sandstones was selected and subjected to freezing tests. The samples were tested after 0 h, 4 h, 8 h, and 12 h, respectively. After the freezing test, nuclear magnetic resonance detection was performed to analyze the *T_2_* distribution. The relationship between the transverse relaxation time *T_2_* of nuclear magnetic resonance and the pore diameter can be expressed by the formula:(45)$$\frac{1}{T_{2}}=\rho \times \frac{S}{V}$$
where *T_2_* is the transverse relaxation time (ms), ρ is the transverse surface relaxation intensity factor, and *S* and *V* are the surface area (cm^2^) and volume (cm^3^) of the pores, respectively.

Yao [36] gave the corresponding relationship between transverse relaxation time *T_2_*, pore throat diameter D, and pore water type according to the mercury injection test and low field-nuclear magnetic resonance test, as shown in Table 5.

Figure 14 shows the pore size distribution of saturated sandstone during freezing. Before freezing, there are only two peaks in the *T_2_* distribution of sandstone, and *T_2_* within the main peak range is less than 2.5 ms, so there is film water in the pores. The *T_2_* in the secondary peak range is in the range of 2.5~100 ms, so the pore has mainly capillary water. *T_2_* > 100 ms belongs to the macropore, and the pore has mainly free water. According to the above pore classification standards, sandstone can be divided into three types of pores: trunk pore, secondary pore, and micropore. In combination with Figure 11, the pores in the sandstone are of the “main branch–side branch” pore structure with poor frost resistance, so the freeze–thaw process is more destructive to the siltstone.

It can be found that the porosity and pore diameter of the frozen samples are larger than those of the unfrozen samples. In general, the first *T_2_* peak area of each sample is the largest, which means that most of the pores in the samples are micropores. It can also be seen that after 4 h, 8 h, and 12 h of freezing treatment, the first *T_2_* peaks of the samples move toward the right side as compared to that of an unfrozen sample (0 h), and the peak area increases as the extension of freezing time, which suggests that the freezing process produces more new pores in these samples. This is also a clear indicator of more severe damage as the extension of freezing time. Macropores have a tendency to expand further, and therefore the maximum relaxation time continues to increase. It is shown that the membrane water in the micropores moves continuously to the freezing secondary pores under freezing conditions, leading to an increase in the secondary pore diameter and the increase in the maximum pore diameter. The water migration path is microporous membrane water → mesoporous capillary water → macroporous free water, which is completely consistent with the migration direction and path of the water migration unit model.

Figure 15 shows the pore size distribution of saturated sandstone during freezing. The proportion of pore distribution under different freezing times can be clearly seen. Then the change rate of the *T_2_* spectral area of sandstone under different freezing times is calculated, as shown in Table 6. With the increase in freezing time, the proportion of pore size change to its internal structure always shows the rule that the growth rate of macropores is greater than that of mesopores and larger than that of small pores. The above results are highly consistent with the fracture propagation model.

## 5. Conclusions

In this paper, to understand the freeze–thaw damage characteristics of sandstone, electron microscope scanning, nuclear magnetic resonance, and uniaxial compression tests were carried out on sandstone with different saturations after freeze–thaw cycles. The microscopic appearance and *T_2_* distribution of nuclear magnetic resonance of sandstones after freeze–thaw cycles were analyzed. Based on the membrane water theory and capillary theory, the water migration path was explored, and the internal fracture diffusion rate of pores with different diameters was analyzed using the Segregation ice theory, and then the evolution law of freeze–thaw damage to sandstone was obtained. The conclusions are as follows:(1)The strength of sandstone decreases first and then increases slowly with the increase in saturation. There are two dividing points in the curve of peak load and damage variable (Figure 5). The frost crack threshold reaches a saturation of 75%, and the saturation threshold reaches a saturation of 85%. When the saturation is before 75%, the peak load loss ratio is 0.409; when the saturation exceeds 75%, the peak load loss ratio rises sharply, reaching 1.356. When the saturation reaches 85%, the peak load reaches the minimum value.(2)The rapid failure of water-bearing rock samples has two causes: the uncoordinated deformation caused by cyclic freeze–thaw causes fatigue damage and expansion stress from a water–ice transition in the freeze–thaw cycles that lead to damage at weak zones in the structure.(3)The porosity of the rock is greatly affected by saturation. When the saturation reaches 85%, the porosity reaches the maximum value of 8.03%, indicating that the internal structure of the sandstone is damaged to the maximum extent. With a further increase in the saturation, the porosity decreases slightly and then tends to be flat. With the increase in saturation, the bound fluid first increases and then decreases to a stable state. In the process of saturation transition from 75% to 85%, the mutual conversion rate of bound fluid and free fluid is the largest. When the saturation reaches 85%, the bound fluid and permeability reach the peak value, and more free fluid is absorbed by more rock particles.(4)With the decrease in pore radius, the freezing temperature also decreases, and the freezing direction of sandstone is consistent with the decreasing direction of pore size. During the freezing process, the migration direction of capillary film water is from micropore → mesopore → macropore.(5)There is a positive correlation between the crack growth rate and the theoretical suction. During the freezing process, the theoretical suction and the theoretical ice pressure increase with the decrease in temperature. The crack growth rate is *V*_macropore_ > *V*_mesopore_ > *V*_micropore_.

## Figures and Tables

**Figure 1 materials-16-02309-f001:**
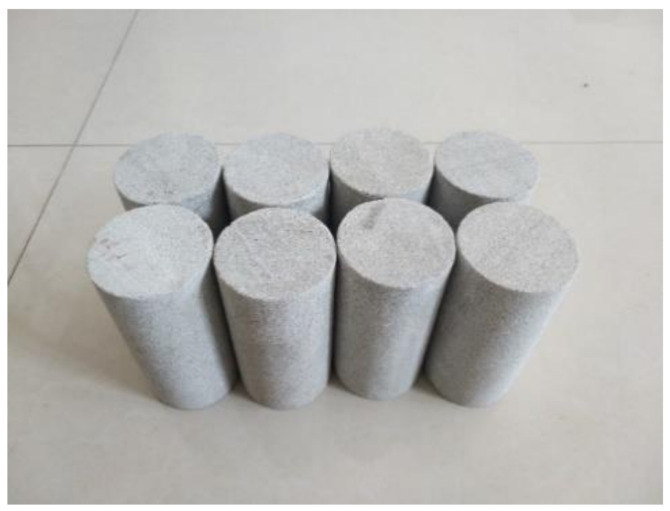
Test sample.

**Figure 2 materials-16-02309-f002:**
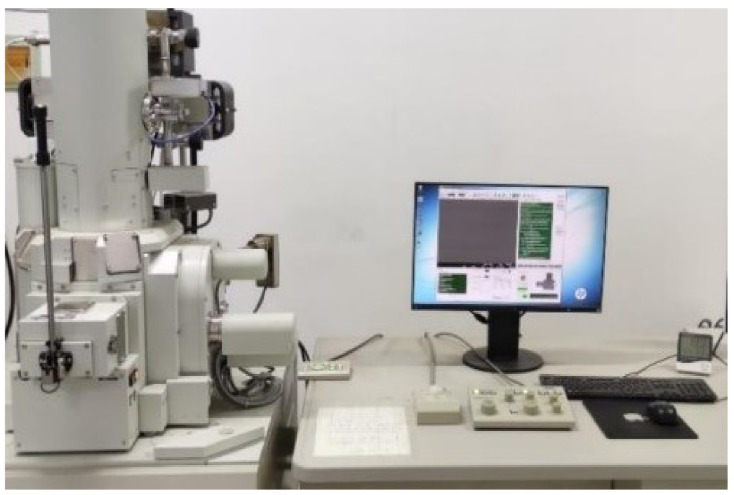
A JSM-7500F SEM.

**Figure 3 materials-16-02309-f003:**
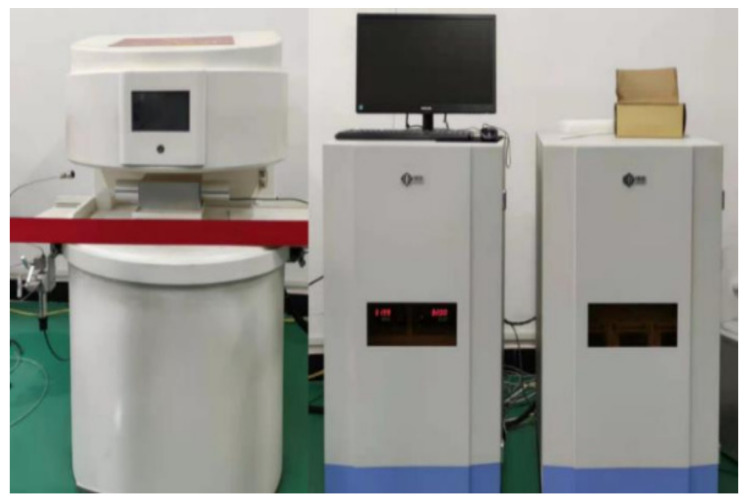
MESOMR 23-060H-I NMR tester.

**Figure 4 materials-16-02309-f004:**
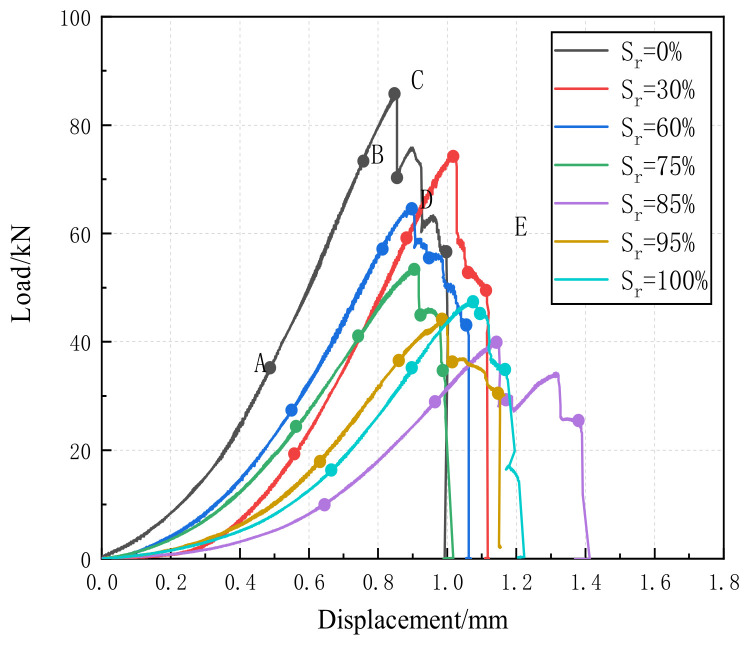
Load displacement curve of sandstone with different saturations.

**Figure 5 materials-16-02309-f005:**
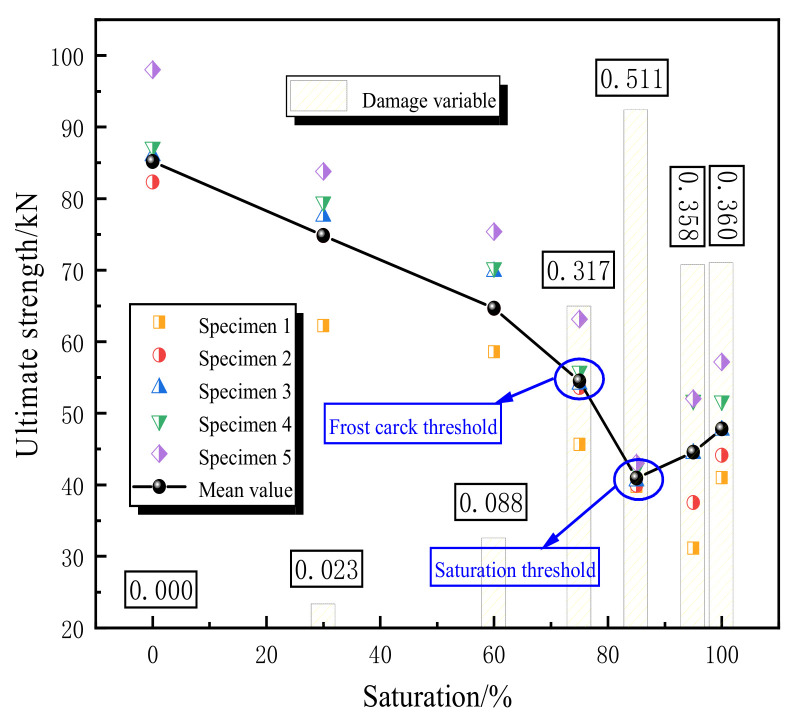
Relationship between peak load and damage variable with saturation.

**Figure 6 materials-16-02309-f006:**
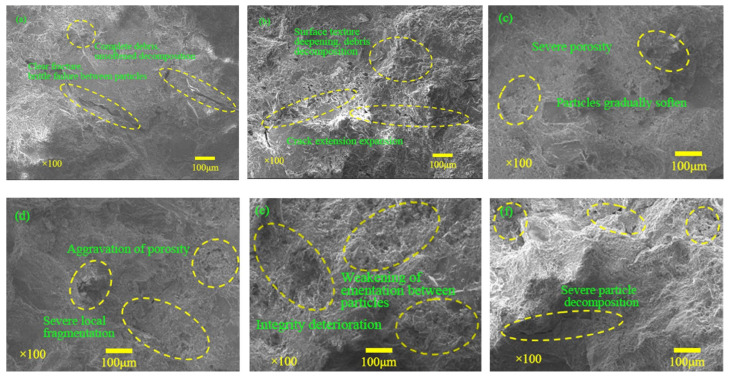
Meso-structural characteristics of sandstone with different saturations under freeze–thaw cycles: (**a**) 0%; (**b**) 30%; (**c**) 60%; (**d**) 75%; (**e**) 95%; (**f**) 100%.

**Figure 7 materials-16-02309-f007:**
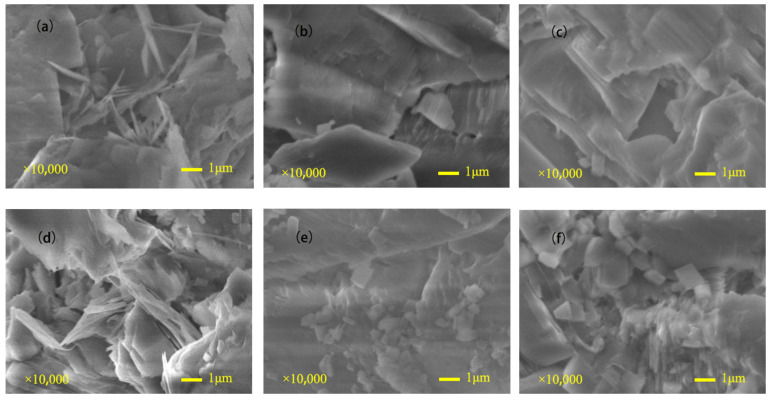
Microscopic appearance of sandstone with different saturation under freeze–thaw cycles: (**a**) 0%; (**b**) 30%; (**c**) 60%; (**d**) 75%; (**e**) 95%; (**f**) 100%.

**Figure 8 materials-16-02309-f008:**
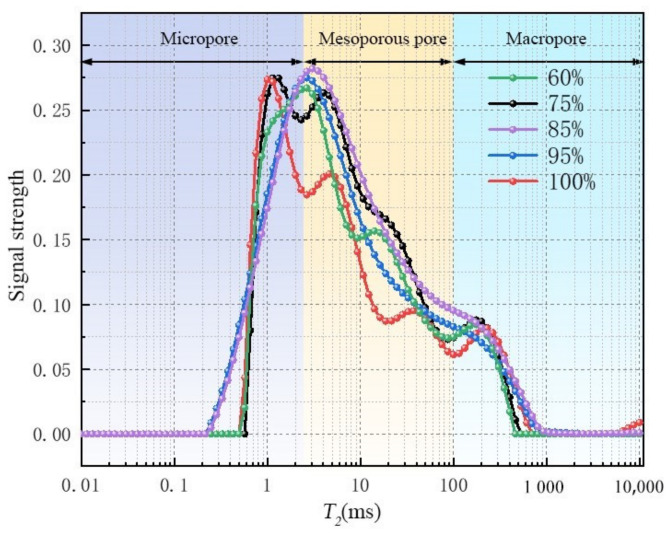
NMR T_2_ distribution of sandstone with different saturation after freeze–thaw.

**Figure 9 materials-16-02309-f009:**
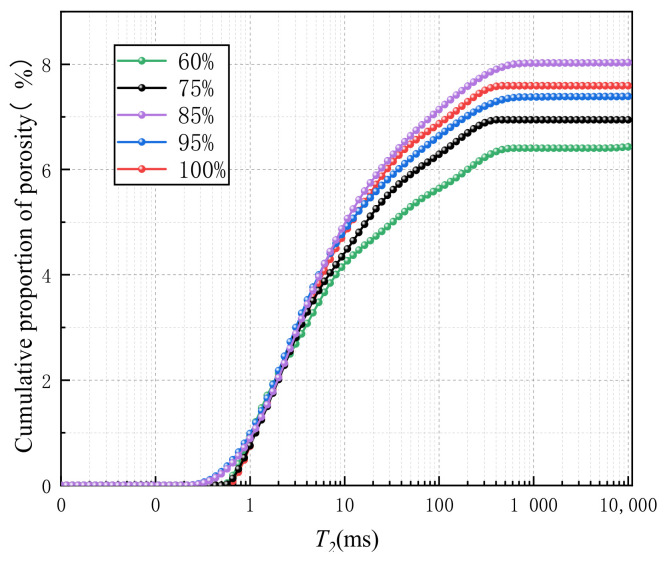
Porosity accumulation curve.

**Figure 10 materials-16-02309-f010:**
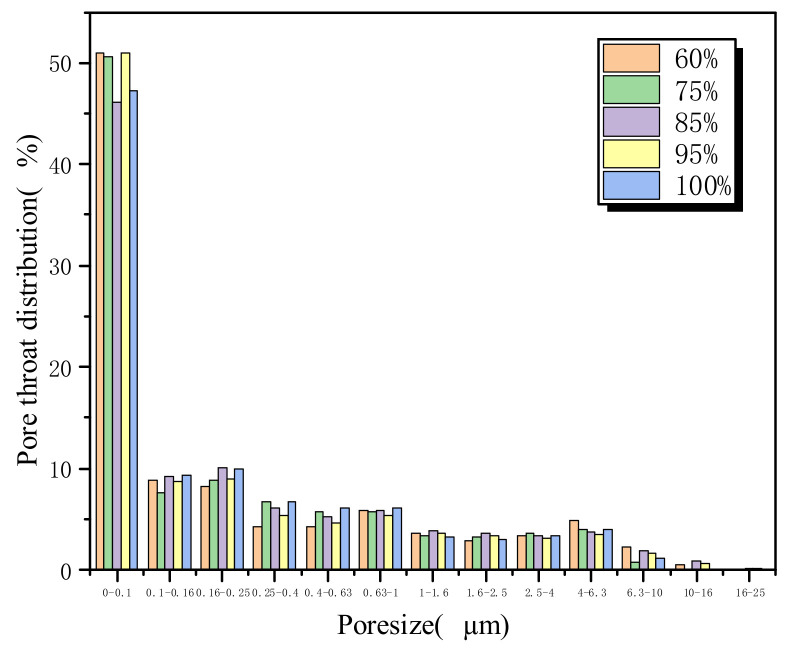
Pore size distribution.

**Figure 11 materials-16-02309-f011:**
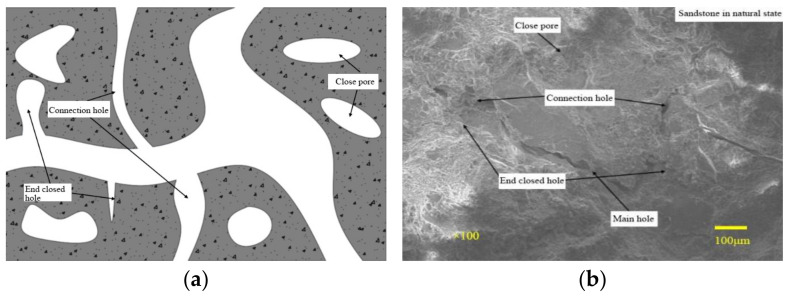
Pore structure in sandstone. (**a**) concept map; (**b**) physical map.

**Figure 12 materials-16-02309-f012:**
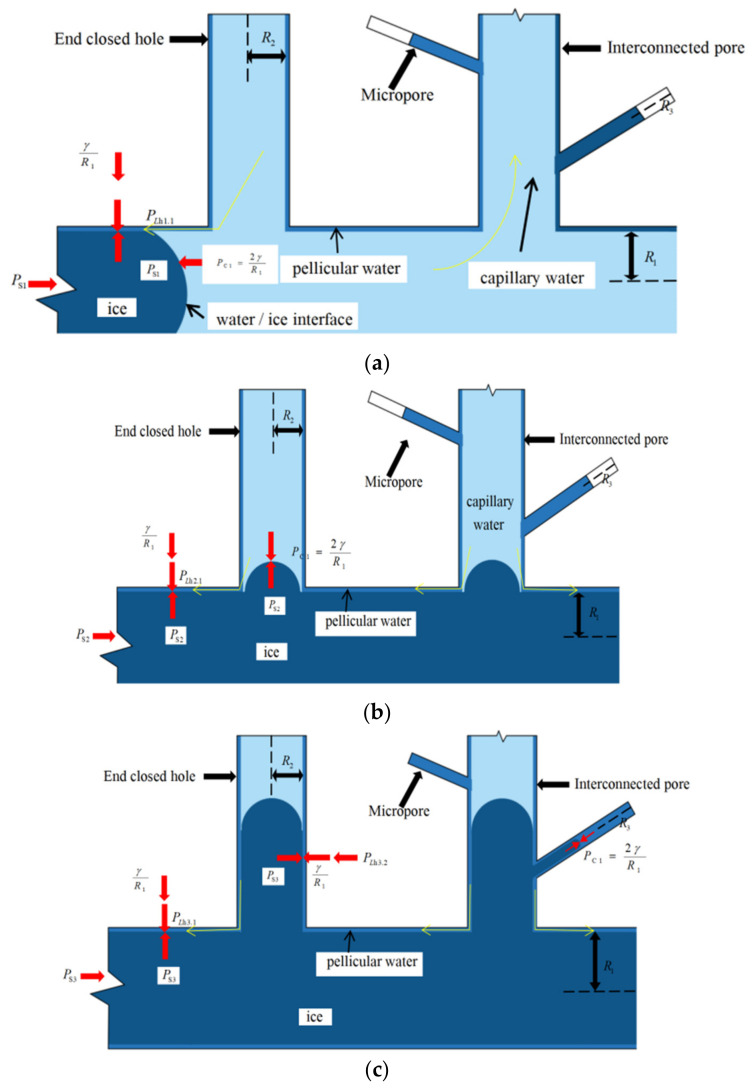
Water transport model. (**a**) macropore freezing; (**b**) mesoporous freezing; (**c**) micropore freezing.

**Figure 13 materials-16-02309-f013:**
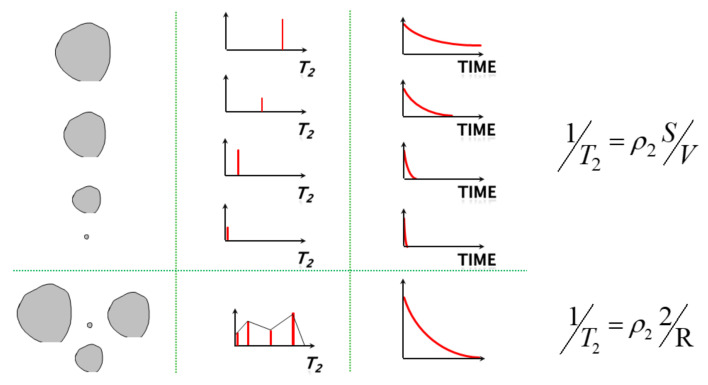
Relaxation time attenuation curve of porous media.

**Figure 14 materials-16-02309-f014:**
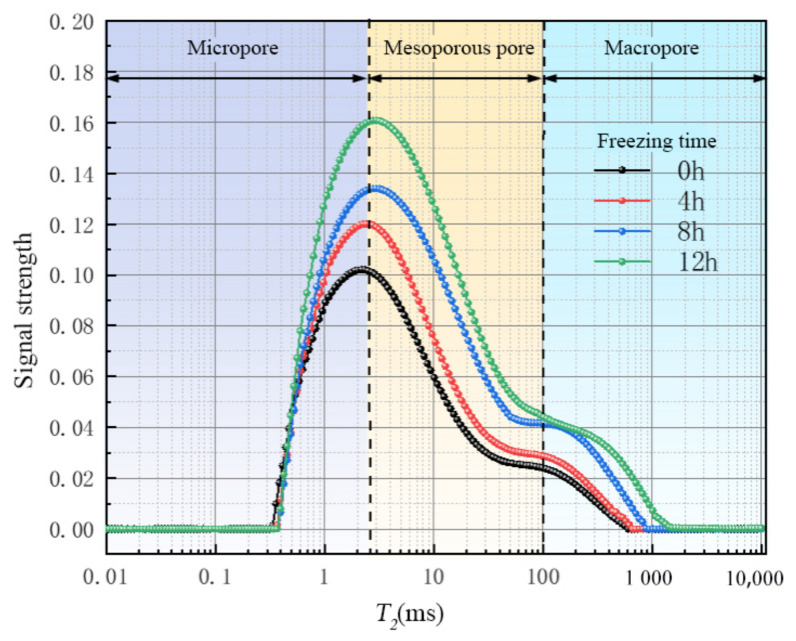
Pore size distribution of saturated sandstone during freezing.

**Figure 15 materials-16-02309-f015:**
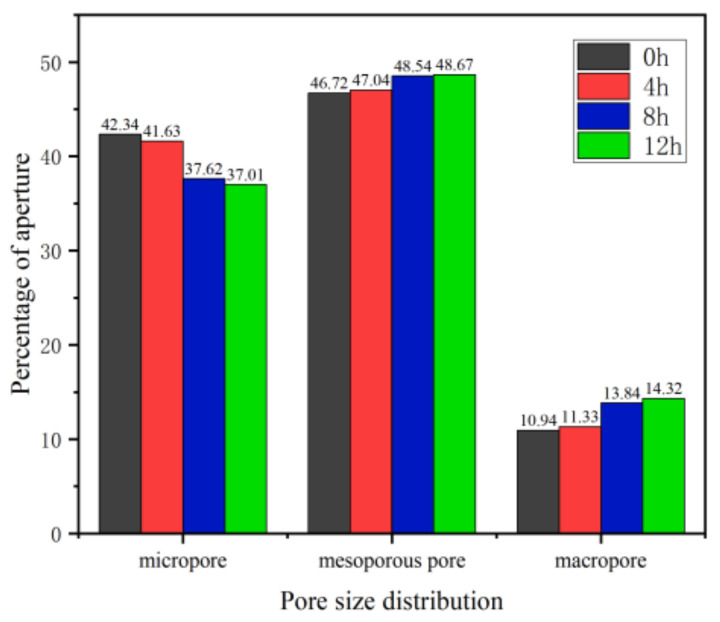
Proportion of pore size of saturated sandstone in the freezing process.

**Table 1 materials-16-02309-t001:** Summary of physical properties of sandstone samples.

Dry Density (g/cm^3^)	Saturated Density (g/cm^3^)	Porosity (%)	Longitudinal Wave Velocity (km/s)
2.33	2.52	9.32	3.07

**Table 2 materials-16-02309-t002:** Uniaxial compression test results.

Saturation (%)	Elastic Modulus (*Mpa*)	Peak Load (*kN*)	Peak Load Loss Rate (%)	Damage Variable
0	68.4	85.17	-	0
30	66.8	74.83	12.1	0.023
60	62.4	64.67	24.1	0.088
75	46.72	54.49	36.0	0.317
85	33.42	40.93	51.9	0.511
95	43.93	44.57	47.7	0.358
100	43.77	47.82	43.9	0.360

**Table 3 materials-16-02309-t003:** NMR parameters of sandstone with different saturation.

Saturation/%	Porosity/%	Bound Fluid/%	Free Flow/%	Permeability/× 10^5^ md
60	6.4284	14.6034	85.3966	45.38
75	6.9411	23.9062	76.0938	58.75
85	8.0285	46.1554	53.8446	136.42
95	7.387	44.2163	55.7837	70.89
100	7.5902	44.0866	55.9134	94.94

**Table 4 materials-16-02309-t004:** Pore classification.

Aperture	Name of Aperture	Classification of Pore Water
<0.1 μm	micropore	thin film-adsorbed water
0.1~1000 μm	mesoporous pore	capillary water
>1000 μm	macropore	gravitational water

**Table 5 materials-16-02309-t005:** Corresponding relation between transverse relaxation time *T_2_* and pore-throat diameter.

$T_{2}$	Aperture	Pore Type	Classification of Pore Water
0.1~2.5 ms	<0.1 μm	absorbed pore	thin film-adsorbed water
2.5~100 ms	>0.1 μm	infiltration pore	capillary water
>100 ms	-	split hole	free water

**Table 6 materials-16-02309-t006:** Aperture ratio change rate (%).

	0 h	4 h	8 h	12 h
micropore	0	−1.68	−9.63	−1.62
mesoporous pore	0	0.68	3.19	0.27
macropore	0	3.56	22.15	3.47

## Data Availability

Data is contained within the article.

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
