# Peer review of "Study on the Influence of Saturation on Freeze–Thaw Damage Characteristics of Sandstone"

_materials, 2023, doi:10.3390/ma16062309_

Round 1
Reviewer 1 Report
In this paper, the authors performed mechanical tests citing uniaxial compression test, electron microscopic scanning and nuclear magnetic resonance to analyze the mechanism of freeze-thaw disaster evolution and the role of water in rock freeze-thaw cycles. The authors actually investigated the law of water migration in the rock, the law of crack growth and the mechanism of damage during freeze-thaw cycles.
In the reviewer opinion, the paper can be recommended for publication in materials journal after addressing the following comments:
- Abstract. Please check the font of the text of line 15
- Authors should highlight the novelty of their manuscript in the abstract
- In the introductory section, the latest publications 2021-2022 should be analyzed more than the oldest references in order to see if the subject remains new or not.
- Line 103-104 the standard reference should be specified
- Lines 114-117 the temperature and drying time of the specimens as well as the saturation pressure and period: are these parameters standardized or arbitrarily chosen? An explanation should be added to the text.
- The authors sometimes use Fig sometimes Figure. The citation of figures should be the same in the whole manuscript.
- Table 1. How is the damage variable noted in the last column calculated?
- Figure 4. The figure legend is unclear; it is difficult to differentiate the curves drawn in this figure.
- The text in Figure 6 is not clear and another color and character should be used.
- Analyzing the same figure 6 what is the better choice or the better saturation condition?
Reviewer 2 Report
The article corresponds to the subject of the journal
The article is written in easy language. Well structured. Written clearly and understandably.
Keywords and abstract are consistent with the conclusions
All references are available
Asa result of the ministry of foreign affairs, itwas established in 1893 in the mountains, when it was carried out in 1893, when it was carried out in 1893. For the purpose ofcatching mines, as well as for the use of mines, as well asfortheconstructionof supports and pipes onthe construction of the Amur River. The first experiment in theartof soil freezing in thecourse of fishing for mines was carried outin 1883 (the orenik "Archibald" in Mawherethe burgokruge, Germania). In the USSR there is a studyof the artsof the land. Freezing of the soil was firstinthe case of menyon in 1928–1929 with theprokhodke kakaliy shah youin the city of SoliKamsk, andthen in 1933 – on the constructionofthe Mosk metroonthe basisof theseme. thathouse proydeny all nacloneshahty for eskalatorov Moskovskogo metro po litena, many iz kotoryh dostigayut glubiny 40-70 m.
Freezing of the soil canbecarried out both in the middleand in the crepe of the nos. Guidesto the geolo gich usli lovi yah. Itis also necessary to buildmines, tonneleu, plotin, dokov, fundamentov zdanii and soon.
The cost of freezing the soil is10–30% of the cost of the ground from the samelevel, with a valueof 40 or more than 40. up to 60% of thetotal costper year.
Freezing of soils, artificial cooling of soils in conditions of natural occurrence to negative temperatures in order to strengthen them and achieve the necessary degree of waterproofness. Therefore, the topic of the authors' research is relevant.
The article contains a detailed literary analysis.
Full-scale experiments, testing were carried out, the physical and mechanical properties of sandstone were determined. Mesostructural characteristics of sandstone of different saturation during freezing-thawing with different cycles are obtained.
Исследования подтверждены electron microscope scanning, nuclear magnetic resonance and uniaxial compression tests
The results are applicable in practice.
Notes: all transcripts or abbreviations should be taken out separately, and removed or deciphered in keywords
Reviewer 3 Report
Review report for the manuscript entitled
“Study on the Influence of Saturation on Freeze Thaw Damage Characteristics of Sandstone”
In this study, the macro mechanical indexes and micro structural characteristics of sandstones with different saturation levels after certain freeze-thaw cycles were analysed using electron microscope scanning, nuclear magnetic resonance, and uniaxial compression tests. The paper is interesting and well-written. However, the authors are required to address the following issues before the manuscript can be accepted for publication:
1) There are some editorial and grammatical mistakes throughout the text. I highly recommend the authors to carefully go through the manuscript and correct these mistakes accordingly. (e.g., “ as the increase of saturation” in the abstract, …)
2) Are the results presented in this study on the macro-meso freeze-thaw damage consistent with the previous studies which have mostly been carried out based on macro-scale properties? The authors are better present a comparison in this regard between their results and those in the previous studies.
3) The literature review is weak. There are a great number of studies on the effect of freeze-thaw cycles on the mechanical properties of soil and rock. I highly suggest the authors to add these studies to the manuscript.
Shirmohammadi, S., Ghaffarpour Jahromi, S., Payan, M., & Senetakis, K. (2021). Effect of lime stabilization and partial clinoptilolite zeolite replacement on the behavior of a silt-sized low-plasticity soil subjected to freezing–thawing cycles. Coatings, 11(8), 994.
Ahmadi, S., Ghasemzadeh, H., & Changizi, F. Effects of A Low-Carbon Emission Additive on Mechanical Properties of Fine-grained Soil under Freeze-Thaw Cycles. Journal of Cleaner Production, 2021, 127157.
Güllü, H., & Khudir, A. Effect of freeze–thaw cycles on unconfined compressive strength of fine-grained soil treated with jute fiber, steel fiber and lime. Cold Regions Science and Technology, 2014, 106, 55-65.
4) Section 2.1: Please summarize the properties of the tested rock in a table.
5) Do the authors have the information of the site where the samples have been taken? Any info on the engineering properties? Rock Classification (RMR & Q)?
6) “, and the porosity was 9.32%.” What is the definition of porosity here? This should be presented.
7) Have the authors followed any standard in applying freeze-thaw cycles (level, temperature, duration, etc.). If so, it should be mentioned.
8) Fig. 4: The legends should be modified. (Sr=…)
9) Fig. 5: Have the saturation threshold corresponding to the minimum degree of saturation been reported in the previous studies? How about the effect of freeze-thaw cycles on the value?
10) Fig. 6: The SEM images should be presented more clearly! The texts are not legible!
11) How can the rock discontinuity orientation affect the results obtained in this study?
12) Table 3: Why do the porosity and permeability experience a peak value at the saturation level of 85%.

Round 2
Reviewer 3 Report
The authors have properly addressed all the raised issues and the manuscript can now be accepted for publication.